# FULLY KOLMOGOROV-ARNOLD DEEP MODEL IN MEDICAL IMAGE SEGMENTATION

## ABSTRACT

Deeply stacked KANs are practically impossible due to high training difficulties and substantial memory requirements. Consequently, existing studies can only incorporate few KAN layers, hindering the comprehensive exploration of KANs. This study overcomes these limitations and introduces the first fully KA-based deep model, demonstrating that KA-based layers can entirely replace traditional architectures in deep learning and achieve superior learning capacity. Specifically, (1) the proposed Share-activation KAN (SaKAN) reformulates Sprecher's variant of Kolmogorov-Arnold representation theorem, which achieves better optimization due to its simplified parameterization and denser training samples, to ease training difficulty, (2) this paper indicates that spline gradients contribute negligibly to training while consuming huge GPU memory, thus proposes the Grad-Free Spline to significantly reduce memory usage and computational overhead. (3) Building on these two innovations, our ALL U-KAN is the first representative implementation of fully KA-based deep model, where the proposed KA and KAonv layers completely replace FC and Conv layers. Extensive evaluations on three medical image segmentation tasks confirm the superiority of the full KA-based architecture compared to partial KA-based and traditional architectures, achieving all higher segmentation accuracy. Compared to directly deeply stacked KAN, ALL U-KAN achieves $10\times$ reduction in parameter count and reduces memory consumption by more than $20\times$, unlocking the new explorations into deep KAN architectures. Code is available at: https://github.com/****.

## 1 INTRODUCTION

Kolmogorov–Arnold Networks (KANs) (Liu et al., 2024c) have emerged as a promising alternative to mainstream architectures such as MLPs and CNNs, and are regarded as a potential high-performance neural network paradigm. Built upon the Kolmogorov–Arnold (KA) representation theorem, KANs employ learnable univariate spline functions as edge-based activation functions. This design enhances their nonlinear expressivity, interpretability, and feature representation capabilities that conventional MLPs with fixed activation functions inherently lack. Empirical evidence has further demonstrated the effectiveness of KANs across diverse applications, including medical image segmentation (Li et al., 2025), image generation (Qiu et al., 2025), and time-series forecasting (Inzirillo & Genet, 2024). These advancements highlight the potential of KANs to serve as a transformative neural network architecture.

Despite their promising advantages, existing studies cannot explore the benefits of deeply stacked KANs, hindered by high training difficulty and excessive memory demands. (1) Training KANs is difficult due to excessive learnable parameters. For identical input–output dimensions, each input dimension in a KAN requires an independent activation function, leading to $O(n_{\text{in}}n_{\text{out}}n_{\text{spline}})$ trainable parameters, which is $n_{\text{spline}}$ times greater than that of a single MLP layer $O(n_{\text{in}}n_{\text{out}})$. (2) Training KANs requires excessive GPU memory. Backpropagation necessitates storing complex gradient maps from spline computations, causing memory usage to scale rapidly with network depth and rendering fully KAN-based architectures impractical for deep networks. Consequently, current research has been restricted to adding few KAN layers into deep models (Li et al., 2025; Cheng et al., 2025; Wang et al., 2025) (Fig. 1) or testing them within shallow networks (Liu et al., 2024c; Bodner et al., 2024; Liu et al., 2024b). This paper breaks these barriers hindering deeper KAN and introduces the

first fully KA-based deep model, demonstrating its superior learning capability in medical image segmentation. This advancement opens a promising direction for the broader applications of KANs.

In this paper, we propose the first fully KA-based deep model. This design breaks the barriers of high training difficulty and excessive memory demands, enabling the scaling of KANs to deep architectures and establishing a foundation for their application in large-scale models. Compared to the vanilla Kolmogorov–Arnold representation theorem, this paper indicates that Sprecher's variant theorem achieves better optimization in deep learning by providing denser training samples to each learnable function and offers a more compact parameterization with equal representational capacity. Specifically, (1) our proposed Shared-activation KAN (SaKAN) reformulates the variant of KA theorem, and eases training difficulty by sharing a single activation function across input dimensions, thereby reducing the number of learnable parameters and increasing the training samples on each activation function. (2) Our proposed Grad-Free Spline strategy significantly reduces memory consumption by detaching spline gradients, while demonstrating negligible impact on learning capacity. (3) Building on these innovations, our ALL U-KAN is the first representative implementation of fully KA-based deep model, in which all fully connected (FC) and convolutional layers are replaced with KA-based layers (KA layers and KAonv) introduced in this paper. It enhances nonlinear learning capacity for high-dimensional and complex tasks. The superior performance of our ALL U-KAN across three medical image segmentation tasks indicates that the fully KA-based deep model possesses stronger learning capabilities than traditional architectures. This advancement will propel the development of a new generation of revolutionary neural network architectures.

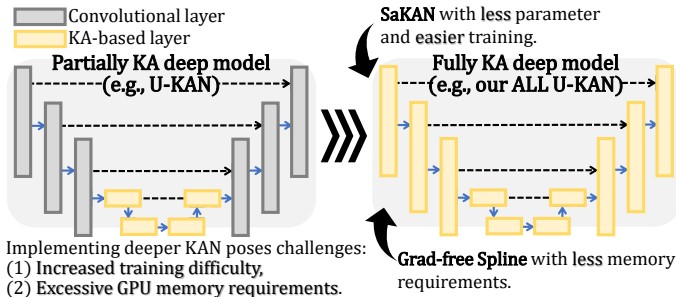

Figure 1: Compared to existing deep methods that are partially based on KAN (e.g., U-KAN), this paper proposes the first deep model composed fully of KA-based layers, exploring its superior performance.

## 2 RELATED WORK

### 2.1 DEEPLY STACKED KANS ARE PRACTICALLY IMPOSSIBLE

Although KANs (Liu et al., 2024c) provide stronger nonlinear feature representation than traditional architectures, deeply stacked KANs are practically impossible due to high training difficulties and substantial memory requirements. This section first reviews KANs, then analyzes these limitations compared to MLPs.

KANs are based on the Kolmogorov–Arnold representation theorem, which states that any multivariate continuous function $f(x)$ can be represented as a finite sum of univariate continuous functions $\phi(x_i)$:

$$f(x_1, \ldots, x_n) = \sum_{i=1}^{2n+1} \Phi_i \Big( \sum_{j=1}^{n} \phi_{ij}(x_j) \Big) \tag{1}$$

Unlike earlier research (Sprecher & Draghici, 2002; Köppen, 2002; Lin & Unbehauen, 1993; Lai & Shen, 2021), KANs adopt modern neural network design principles by stacking deeper, producing smoother activation functions, more stable training, and greater representational capacity (Liu et al., 2024c). For each KAN layer, given inputs $[x_1, \ldots, x_{n_{in}}]$ and outputs $[y_1, \ldots, y_{n_{out}}]$, the layer is formally expressed as:

$$\mathbf{Y} = [y_1, \ldots, y_{n_{out}}] = \text{KAN}([x_1, \ldots, x_{n_{in}}]) \tag{2}$$

KAN defines a set of learnable 1D functions:

$$\Phi = \{\phi_{q,p}\}, \quad q = 1, \ldots, n_{out}, \ p = 1, \ldots, n_{in} \tag{3}$$

Each output $y_i$ is represented as a weighted sum of univariate functions $\phi_{ij}(x_j)$, where each $\phi_{ij}$ is modeled using B-spline functions. Let $n_{\text{spline}}$ denote the number of spline basis functions. Then the formulation is:

$$y_i = \sum_{j=1}^{n_{\text{in}}} \left( \phi_{ij}(x_j) + u_{ij}\, b(x_j) \right) = \sum_{j=1}^{n_{\text{in}}} \left( \sum_{k=1}^{n_{\text{spline}}} v_{ijk}\, B_k(x_j) + u_{ij}\, b(x_j) \right) \tag{4}$$

where $b(\cdot)$ is a linear residual (defaulting to SiLU) introduced to improve training stability and $B(\cdot)$ denotes spline basis functions. Both $b(\cdot)$ and $B(\cdot)$ are fixed and parameter-free, while $u_{ij}$ and $v_{ijk}$ are trainable parameters. Consequently, the entire KAN layer can be expressed compactly in matrix form as:

$$\mathbf{Y} = \underbrace{\begin{pmatrix} \phi_{1,1}(\cdot) & \cdots & \phi_{1,n_{\text{in}}}(\cdot) \\ \vdots & \ddots & \vdots \\ \phi_{n_{\text{out}},1}(\cdot) & \cdots & \phi_{n_{\text{out}},n_{\text{in}}}(\cdot) \end{pmatrix}}_{\Phi} \mathbf{X} + \underbrace{\begin{pmatrix} u_{1,1} & \cdots & u_{1,n_{\text{in}}} \\ \vdots & \ddots & \vdots \\ u_{n_{\text{out}},1} & \cdots & u_{n_{\text{out}},n_{\text{in}}} \end{pmatrix} b(\mathbf{X})}_{\text{Linear Residuals}} \tag{5}$$

To investigate why deeply stacked KANs are impossible in practical applications, we analyze their limitations in terms of training difficulties and memory requirements using MLPs as the baseline.

**KAN is more difficult to train** due to its more training parameters and fewer available samples. A single KAN layer requires $O(n_{\text{in}} n_{\text{out}} n_{\text{spline}})$ trainable parameters, which is $n_{\text{spline}} \times$ more than an FC layer with only $O(n_{\text{in}} n_{\text{out}})$. Under the default configuration (grid size = 5, order = 3), the $n_{\text{spline}} = 8$. Since KAN assigns an independent activation function to each input dimension, the available training samples for each parameter are limited to a single input dimension, whereas in MLPs, each parameter is trained on all input dimensions.

**Training deep KANs requires substantial memory resources.** Compared with conventional architectures, KANs involve significantly more learnable parameters and rely on the computationally intensive de Boor–Cox formula (Cox, 1972; De Boor & De Boor, 1978) to construct spline basis $B_k(\cdot)$ in Eq. 4. Consequently, training necessitates the storage of highly complex gradient graphs for backpropagation, which results in excessive memory consumption and reduced training speed.

Notably, although prior studies (Wang et al., 2025; Bodner et al., 2024; Liu et al., 2024c) indicate that KANs achieve superior performance with fewer *overall* parameters, owing to stronger nonlinear representation capacity compared to deeper MLPs, Our focus is on scaling KANs to deeper. At equivalent scale, KANs do require more trainable parameters.

These limitations hinder the extension of KANs to deeper architectures, preventing them from fully replacing conventional structures in deep models and thereby constraining their broader development and practical applications.

## 2.2 Existing Methods cannot exploit deep KAN

The training difficulty and memory requirements of deeper KANs increase dramatically as shown in Sec. 2.1, hindering the exploration of their potential to fully replace traditional architectures and constraining further development. As a result, current research remains limited to shallow KANs or deep models that incorporate only a few KAN layers, providing only a partial assessment of their capabilities.

Existing fully KAN-based methods cannot be scaled to deeper and remain restricted to simple tasks. Early research explored the feasibility of constructing novel networks based on the Kolmogorov-Arnold (KA) theorem (Sprecher & Draghici, 2002; Köppen, 2002; Lin & Unbehauen, 1993; Lai & Shen, 2021; Leni et al., 2011; Fakhoury et al., 2022; Montanelli & Yang, 2020), but most of them adhered to the two-layer, $2n+1$ width formulation. KANs (Liu et al., 2024c) aim to extend the KA theorem to arbitrary widths and depths, yet high training costs and resource requirements limited them to only $3-5$ layers and simple tasks such as function approximation and MNIST classification. KAN 2.0 (Liu et al., 2024b) introduced multiplicative nodes and scientific knowledge embedding to showcase shallow KANs for scientific discovery. Convolutional KANs (Bodner et al., 2024) incorporated convolutional structures, surpassing CNNs on Fashion-MNIST with $3-5$ layer

KAN convolutional networks. LDSB (Qiu et al., 2025) leveraged a single-layer KAN to model diffusion path weights in generative diffusion models. SigKAN (Inzirillo & Genet, 2024) and GKAN (Kiamari et al., 2024) demonstrated that shallow KANs improve performance in time-series forecasting and graph data processing.

For more complex tasks, existing methods improve learning by incorporating few KAN layers within deep models. U-KAN (Li et al., 2025) and Implicit U-KAN2. 0 (Cheng et al., 2025) integrated tokenized KAN and MultiKAN blocks, respectively, into the U-Net architectures, enhancing medical image segmentation. LoKi (Cai et al., 2025) introduced a shallow hybrid of FC and KAN layers for parameter-efficient fine-tuning, mitigating overfitting in downstream tasks. MedKAFormer (Wang et al., 2025) embedded KAN layers at critical stages of ViT pipelines to strengthen medical image representations.

Despite these advances, challenges in deep stacking KANs still limit their exploration in deep learning. This work overcomes these challenges and introduces the first fully KA-based deep model, demonstrating the advantages of replacing all FC and conv layers with KA-based layers for complex tasks and establishing a foundation for future research.

## 3 Methods

This paper overcomes key computational barriers to make deep stacked KANs practically feasible, and proposes the first fully KA-based deep model. Specifically, the Shared-activation KAN (SaKAN, Sec. 3.1) is proposed to ease the training difficulty of deep KANs based on the Sprecher's variant of KA theorem, the Grad-Free Spline (Sec. 3.2) is introduced to reduce memory consumption during training. Building on these advancements, ALL U-KAN (Sec. 3.2) is the first representative implementation of fully KA-based deep network, outperforming traditional architectures.

### 3.1 Shared-activation KAN eases training difficulty

Based on Sprecher's variant of the KA theorem, the Shared-Activation KAN (SaKAN) is proposed to ease the training difficulty while maintaining the powerful representation.

#### 3.1.1 Variants of the Kolmogorov–Arnold Representation Theorem

This section indicates that the Sprecher's variant Sprecher (1965) of KA theorem simplifies parameterization and enhances optimization by providing more training samples to each learnable function, overcoming the high training complexity and poor scalability of vanilla KA theorem Eq. 1, which forces KANs to learn independent activation functions $\phi_{ij}$ for each input dimension.

In Sprecher (1965) version, each independent $\phi_{ij}$ in Eq. 1 is replaced with a unified function $\phi$, with input variables appropriately shifted. This modification greatly simplifies the parameterization by reducing the number of activation functions for training. Specifically, for a multivariate continuous function $f : [0, 1]^n \to \mathbb{R}$, there exist real values $\eta, \lambda_1, \ldots, \lambda_n$, a continuous function $\Phi : \mathbb{R} \to \mathbb{R}$, and an real monotonie increasing function $\phi \in \mathrm{Lip}\,(\ln 2/\ln(2N + 2))$, where $N \geq n \geq 2$:

$$f(\mathbf{x}) = \sum_{i=1}^{2n+1} \Phi\left( \sum_{j=1}^{n} \lambda_j\, \phi(x_j + \eta i) + i \right) \tag{6}$$

This variant offers distinct advantages over the vanilla KA theorem for deep learning: (1) **Simpler parameterization.** The KA theorem requires $(n + 1)(2n + 1)$ nonlinear functions, whereas the variant needs only 2 functions and $n+1$ real constants to represent the same continuous function. (2) **More training samples for each learnable function.** In the vanilla formulation, each function $\phi_{ij}$ is sampled only from $x_j$. In contrast, the variant employs a unified function $\phi$ that is sampled jointly from all inputs $x_1, \ldots, x_n$, thereby enabling more effective learning. Together, these advantages highlight the suitability of Sprecher's variant in deep learning.

#### 3.1.2 SaKAN reformulates the Variant Theorem

The SaKAN reformulates the Sprecher's variant of the KA theorem, and shares a single activation function across input dimensions, with fewer trainable parameters and more training samples per

function. Consequently, SaKAN eases training difficulty while preserving the strong representational capacity of KANs.

For an input vector $\mathbf{X} = [x_1, \ldots, x_{n_{\text{in}}}]$ and an output vector $\mathbf{Y} = [y_1, \ldots, y_{n_{\text{out}}}]$, the SaKAN layer is defined as $\mathbf{Y} = \text{SaKAN}(\mathbf{X})$. For each output $y_i$, the formulation is:

$$
y_i = \sum_{j=1}^{n_{\text{in}}} \Big( \phi_i(x_j) + u_{ij}\, b(x_j) \Big) = \sum_{j=1}^{n_{\text{in}}} \left( \sum_{k=1}^{n_{\text{spline}}} v_{ik}\, B_k(x_j) + u_{ij}\, b(x_j) \right)
$$
$$
= \sum_{k=1}^{n_{\text{spline}}} v_{ik} \sum_{j=1}^{n_{\text{in}}} B_k(x_j) + \sum_{j=1}^{n_{\text{in}}} u_{ij}\, b(x_j)
\tag{7}
$$

The formulation can be further expressed in matrix form:

$$
\mathbf{Y} = \underbrace{\begin{pmatrix} \mathbf{1}_{n_{\text{in}}}\, \phi_1(\mathbf{X}) \\ \vdots \\ \mathbf{1}_{n_{\text{in}}}\, \phi_{n_{\text{out}}}(\mathbf{X}) \end{pmatrix}}_{\Phi} + \underbrace{\begin{pmatrix} u_{1,1} & \cdots & u_{1,n_{\text{in}}} \\ \vdots & \ddots & \vdots \\ u_{n_{\text{out}},1} & \cdots & u_{n_{\text{out}},n_{\text{in}}} \end{pmatrix} b(\mathbf{X})}_{\text{Linear Residuals}}
\tag{8}
$$

where $\mathbf{1}_{n_{\text{in}}} := [1, \ldots, 1] \in \mathbb{R}^{1 \times n_{\text{in}}}$ and $\mathbf{X} \in \mathbb{R}^{n_{\text{in}} \times 1}$.

Compared to vanilla KAN (Eq. 5), SaKAN can be regarded as a simplified version, while maintaining the equivalent representational capacity, as demonstrated by the KA variants in Sec. 3.1.1. SaKAN reduces the number of nonlinear learnable functions $\phi$ from $n_{\text{in}} \times n_{\text{out}}$ to $n_{\text{out}}$, reducing training parameters. By sharing each $\phi$ across all input dimensions rather than a single dimension, SaKAN increases the number of training samples per function, as illustrated in Fig. 2 and Eq. 8.

Compared with Sprecher's variant of the KA theorem, SaKAN introduces three key modifications that enhance generalization and representational power: (1) SaKAN replaces the shifted unified function $\phi(x_j + \mu i)$ in Eq. 6 with independent functions $\phi_i(x_j)$. (2) SaKAN introduces linear residuals $u_{ij}, b(x_j)$ to represent the additive term $i$ in Eq. 6. The (1) and (2) extend the representational scope to

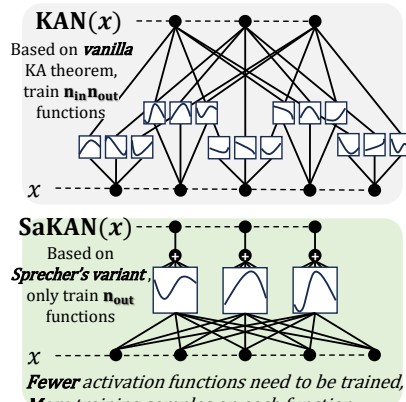

Figure 2: KAN vs. our SaKAN

fully cover the original variant. (3) SaKAN simplifies all coefficients $\lambda_j$ to 1, which can be offset by deeper stacking. This modification not only preserves the order sensitivity of $\phi$, as its linear residual remains responsive to input order, but also empirically mitigates overfitting, reduces gradient-map complexity, and significantly decreases memory consumption. These adaptations enable SaKAN to retain the theoretical strengths of the KA variant while making it substantially more scalable and efficient for deep learning applications.

## 3.2 GRAD-FREE SPLINE REDUCE MEMORY USAGE

The proposed Grad-Free Spline detaches the spline gradients during training as shown in Fig. 3a, to overcome a key challenge in stacking KANs deeper, namely the substantial memory overhead caused by storing complex gradient graphs. Theorem 1 demonstrates that it maintains both training efficiency and representational capacity.

**Theorem 1** (Grad-Free Spline Preserves Optimization). *Using Grad-Free Spline at layer $L$ has minimal impact on the optimization of current or preceding layers $L - 1$. Let layer $L$ of a KAN have learnable weights $u^L$ and $v^L$, with input $\mathbf{x}^L$ and output $\mathbf{y}$, and previous layer input $\mathbf{x}^{L-1}$, expressed as $\mathbf{x}^{L-1} \to \mathbf{x}^L \to \mathbf{y}$.*

*Proof.* Step 1: Grad-free Spline does not alter the gradients of $v^L$ and $u^L$ in the current layer $L$. According to forward process Eq. 7, the gradients of $u^L$ and $v^L$ are

$$\frac{\partial y_i}{\partial v_{ik}^L} = \sum_{j=1}^{n_{\text{in}}} B_k(x_j^L), \quad \frac{\partial y_i}{\partial u_{ij}^L} = b(x_j^L) \tag{9}$$

which depend only on the spline bases $B_k(\cdot)$, not on their derivatives. Thus, detaching spline gradients does not affect the optimization of $u^L$ and $v^L$.

Step 2: It does not hinder the optimization of preceding-layer $L-1$ gradients $v^{L-1}$ and $u^{L-1}$. For layer $L-1$, the gradient of its parameters is

$$\frac{\partial y}{\partial v_{ik}^{L-1}} = \frac{\partial y}{\partial \mathbf{x}^L} \frac{\partial \mathbf{x}^L}{\partial v_{ik}^{L-1}}, \quad \frac{\partial y}{\partial u_{ij}^{L-1}} = \frac{\partial y}{\partial \mathbf{x}^L} \frac{\partial \mathbf{x}^L}{\partial u_{ij}^{L-1}} \tag{10}$$

where the first term $\frac{\partial y}{\partial \mathbf{x}^L}$ is the layer-$L-1$ output gradient, and the second term follows the same form as Eq. 9. The gradient of $\mathbf{x}^L$ decomposes as

$$\frac{\partial y}{\partial \mathbf{x}^L} = \underbrace{\sum_{k=1}^{n_{\text{spline}}} v_{ik}^L \sum_{j=1}^{n_{\text{in}}} B_k'(x_j^L)}_{G_S} + \underbrace{\sum_{j=1}^{n_{\text{in}}} u_{ij}^L \, b'(x_j^L)}_{G_L} \tag{11}$$

where $G_S$ is propagated through the B-**S**pline, and $G_L$ is propagated through the **L**inear residual connection. Notably, the gradient magnitude of $G_S$ is significantly smaller than that of $G_L$, as:

$$\left| \frac{\partial y}{\partial \mathbf{x}^L} \right| \approx |G_L| \gg |G_S| \tag{12}$$

indicating that spline gradients $G_S$ have negligible effect on dominant gradient propagation, because: (a) Local support and Gradient sparsity. Each spline basis has compact support, so only a few derivatives $B_k'(x)$ are nonzero for a given input. In contrast, the $b'(x)$ (defaulting to SILU) is nonzero almost everywhere, leading to a dense gradient. (b) Cancellation effect. The $B_k'(x)$ takes both positive and negative values, and these values are symmetric, with $\int B_k'(x)dx = 0$. When aggregated across inputs, these opposing contributions may cancel out. In contrast, the positive values of $b'(x)$ (SiLU) exceed the negative values, leading to constructive accumulation.

This theoretical observation is also supported by empirical evidence from small-scale KANs (with dimensions $(4, 128, 128, 1)$ and random inputs), as shown in Fig. 3b.

By induction, this argument extends to all preceding layers, proving that Grad-Free Spline preserves stable training in deep KANs. $\square$

After detaching gradients, memory usage can be further reduced via chunk-wise computation of the grad-free spline basis. The trade-off between chunk size and runtime efficiency is analyzed in ablation experiments.

### 3.3 ALL U-KAN

Based on SaKAN (Sec. 3.1) and Grad-Free Spline (Sec. 3.2), ALL U-KAN is the first representative implementation of a fully KA-based deep model, demonstrating that our KA-based layers can entirely replace traditional layers.

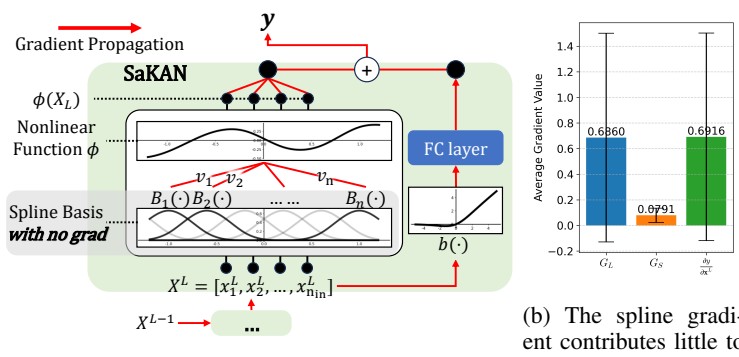

(a) Grad-Free Spline.

(b) The spline gradient contributes little to training.

Figure 3: Unlike vanilla KANs, Grad-Free Spline shows that detaching spline basis gradients maintains training stability, reduces computational memory, and improves efficiency.

Inspired by KAN (Liu et al., 2024c) and KANConv (Bodner et al., 2024), we propose the KA and KAonv layers. The KA layer replaces the FC layer by applying Grad-Free Spline to the SaKAN. Its parameter complexity is $\mathcal{O}(n_{\text{in}}n_{\text{out}} + n_{\text{out}}n_{\text{spline}})$, which is in same order as $\mathcal{O}(n_{\text{in}}n_{\text{out}})$ of FC layer, and is significantly smaller than $\mathcal{O}(n_{\text{in}}n_{\text{out}}n_{\text{spline}})$ of vanilla KAN layer. In this paper, $n_{\text{spline}} = 8$ is equal to the grid size of 5 plus the spline order of 3. The KAConv layer replaces the con-volutional layer by performing con-volution with kernels, unfolding the values within each receptive field, and outputting the results through a KA layer.

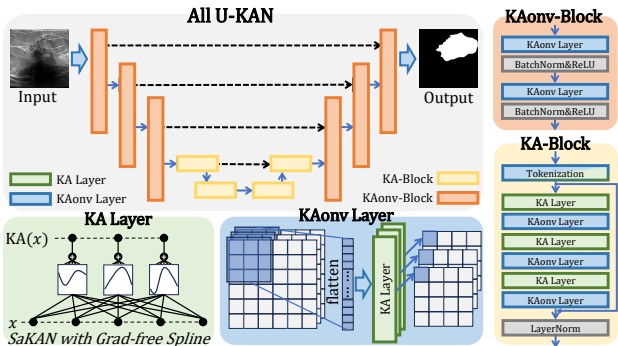

Our ALL U-KAN replaces all FC and KAN layers in U-KAN (Li et al., 2025) with KA layers and substi-tutes every convolutional layer with KAonv layers, resulting in a fully KA-based deep architecture compris-ing 31 KAonv layers and 12 KA lay-ers (details in Fig. 4). This de-sign establishes the first deep model built entirely from KA-based com-ponents while keeping the parameter count and memory usage comparable to those of conventional architectures. In this way, it effectively overcomes the long-standing limi-tations that previously prevented KAN models from scaling to deep networks.

Figure 4: Overview of ALL U-KAN. The proposed KA and KAonv layers, built on SaKAN and Grad-free Splines, re-place all FC and convolutional layers, yielding the first fully KA-based deep model.

## 4 EXPERIMENTS

In this section, our proposed ALL U-KAN demonstrates superior performance across all three medi-cal image segmentation tasks, demonstrating that deeper KA-based networks possess superior learn-ing capabilities over traditional architectures in complex tasks.

### 4.1 DATASETS

There are three medical image segmentation datasets used to comprehensively evaluate the effec-tiveness of our methods.

**BUSI dataset** (Al-Dhabyani et al., 2020): This dataset contains ultrasound images of normal, be-nign, and malignant breast cancer cases, along with corresponding segmentation masks. A total of 647 benign and malignant breast tumor images were used, with all images resized to $256 \times 256$. **GlaS dataset** (Valanarasu et al., 2021): This dataset consists of 612 standard-definition (SD) frames ($384 \times 288$) from 31 sequences collected from 23 patients at the Hospital Clinic of Barcelona. Fol-lowing common practice (Liu et al., 2024a; Li et al., 2025), 165 images were selected and resized to $512 \times 512$. **CVC-ClinicDB** (Bernal et al., 2015): This public dataset is designed for polyp detection in colonoscopy videos. It comprises 612 images ($384 \times 288$) from 31 colonoscopy sequences, with all images uniformly resized to $256 \times 256$.

All datasets were randomly divided into 80% training and 20% validation sets, consistent with the U-KAN Li et al. (2025). Results are based on 3 random runs with the random seed 2981, 6142, and 1187.

### 4.2 EXPERIMENTAL SETTINGS

All experiments were implemented on a single NVIDIA RTX 3090 GPU. For every dataset, the batch size was set to 8, with an initial learning rate of $1e-4$, optimized using Adam with a cosine annealing learning rate scheduler (minimum $1e-5$). The loss function combined Binary Cross-Entropy (BCE) and Dice loss, and all models were trained for 400 epochs.

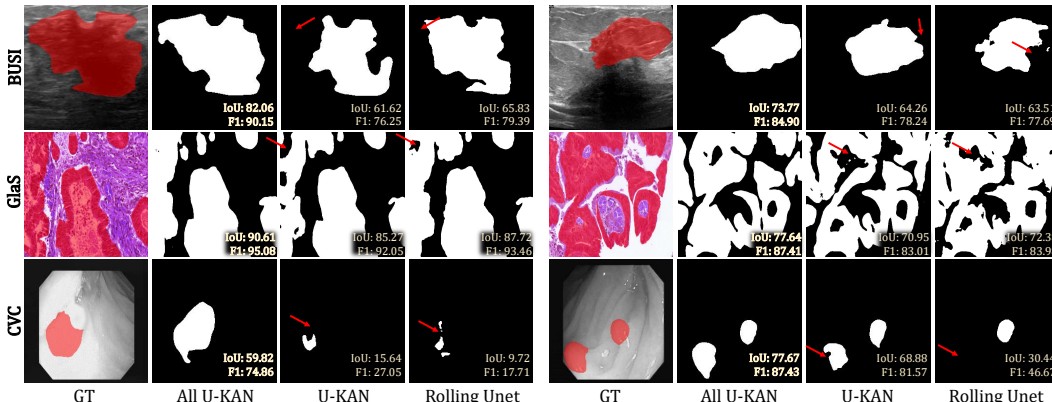

Figure 5: Qualitative comparison. Our ALL U-KAN demonstrates superior segmentation performance overall.

To validate the effectiveness of the proposed method, comprehensive comparisons were performed against state-of-the-art U-shaped architectures for medical image segmentation. The baselines included the convolution-based U-Net (Ronneberger et al., 2015), U-Net++ (Zhou et al., 2018), the attention-based Att-UNet (Oktay et al., 2018), the Mamba-based U-Mamba (Ma et al., 2024), the advanced MLP-based U-NeXt (Valanarasu & Patel, 2022), Rolling-UNet (Liu et al., 2024a), as well as the KAN-incorporated U-KAN (Li et al., 2025).

Evaluation metrics covered segmentation accuracy (Intersection-over-Union (IoU) and F1 score), network size (parameter count), and training efficiency (GPU memory consumption and runtime). This comprehensive evaluation ensures a rigorous assessment of both performance and efficiency across all competing models.

### 4.3 QUALITATIVE AND QUANTITATIVE COMPARISON

Our fully KA-based deep model achieves superior segmentation performance. Tab. 1 and 2 demonstrate that our All U-KAN achieves substantial improvements across all three datasets. Benefiting from the stronger feature learning capability provided by more KA-based layers, our model significantly outperforms U-KAN in the average results across three data splits, with IoU increases of 2.76, 1.08, and 0.32, and F1-score improvements of 2.24, 0.63, and 0.10. In particular, on the BUSI dataset with the random split seed set to 2981, the IoU and F1 scores improve by 4.17 and 3.37, respectively, as shown in Tab. 2. Figure 5 illustrates that our model's predictions are closer to the ground truth and maintain higher accuracy even when other models fail to segment correctly, further confirming that a deep model composed entirely of KA layers can achieve markedly superior segmentation performance.

### 4.4 ABLATION STUDY

This section evaluates each proposed component as shown in Tab. 2: (1) *SaKAN reduces parameters while maintaining learning capability*. The ⓒ achieves comparable performance to ⓑ with 57% fewer parameters, and ⓙ outperforms ⓖ using only 10% parameters. (2) *Grad-free Spline lowers memory usage with minimal performance loss*. ⓓ outperforms ⓐ while slightly inferior to ⓑ, indicating that KANs without spline gradients still outperform MLPs. ⓔ outperforms ⓓ, shows SaKAN exhibits greater robustness to spline gradients. For deeper stacks, this paper compared memory requirements of ⓕ and ⓗ on batch size of 1, as they exceeded the device capacity, ⓕ required 12.19 GB, ⓖ required 10.48 GB, whereas our ⓙ consumed only 0.597 GB, representing a reduction of more than 20×. (3) *SaKAN + Grad-free Spline scale KANs deeper*. Without them, ⓕ and ⓗ cannot train, and ⓖ uses far more memory than ⓐ and ⓑ. (4) *Deep KANs improve performance*. Fully KA-based ⓙ outperforms both partial-KAN ⓑ and MLP ⓐ, while ⓖ surpasses ⓓ. (5) *Weights $\lambda_j$ in Eq. 6 are unnecessary*. ⓘ underperforms ⓙ, tends to overfit, and increases memory, so SaKAN adopts a weight-free design, and confirms the statement in Sec. 3.1.2.

Table 1: Comparison with state-of-the-art methods on BUSI, GlaS, and CVC datasets. Results are based on three fixed-seed random data splits.

| Methods | BUSI | | GlaS | | CVC | |
|---|---|---|---|---|---|---|
| | IoU↑ | F1↑ | IoU↑ | F1↑ | IoU↑ | F1↑ |
| U-Net | 57.22±4.74 | 71.91±3.54 | 86.66±0.91 | 92.79±0.56 | 83.79±0.77 | 91.06±0.47 |
| Att-Unet | 55.18±3.61 | 70.22±2.88 | 86.84±1.19 | 92.89±0.65 | 84.52±0.51 | 91.46±0.25 |
| U-Net++ | 57.41±4.77 | 72.11±3.90 | 87.07±0.76 | 92.96±0.44 | 86.11±1.47 | 91.53±0.88 |
| U-NeXt | 59.06±1.03 | 73.08±1.32 | 84.51±0.37 | 91.55±0.23 | 74.83±2.44 | 85.36±0.17 |
| Rolling-UNet | 61.00±0.64 | 74.67±1.24 | 86.42±0.96 | 92.63±0.62 | 82.87±1.42 | 90.48±0.83 |
| U-Mamba | 61.81±3.24 | 75.55±3.01 | 87.01±0.39 | 93.02±0.24 | 84.79±0.58 | 91.63±0.39 |
| U-KAN | 63.38±2.83 | 76.40±2.90 | 87.64±0.32 | 93.37±0.16 | 85.05±0.53 | 91.88±0.29 |
| ALL U-KAN | **66.14±5.43** | **78.64±5.33** | **88.72±0.23** | **94.00±0.07** | **85.37±0.06** | **91.98±0.02** |

Table 2: Ablation study on each proposed component. ✓ denotes the inclusion of a component. "U-KAN$_{MLP}$" corresponds to U-KAN with all KAN layers replaced by MLP. The random data splitting seed is 2981, with a batch size of 8 on the BUSI dataset.

| Methods | SaKAN | Grad-free | KAonv | IoU ↑ | F1 ↑ | Params ↓ | GPU Mem. ↓ |
|---|---|---|---|---|---|---|---|
| ⓐ U-KAN$_{MLP}$ | – | – | – | 63.49 | 77.07 | 2.76M | 2.49GB |
| ⓑ U-KAN | – | – | – | 65.26 | 78.75 | 6.35M | 2.52GB |
| ⓒ | ✓ | – | – | 65.16 | 78.60 | 2.78M | 2.49GB |
| ⓓ | – | ✓ | – | 64.46 | 77.94 | 6.35M | 2.51GB |
| ⓔ | ✓ | ✓ | – | 64.74 | 78.24 | 2.78M | 2.49GB |
| ⓕ | – | – | ✓ | – | – | 30.34M | ≫24GB |
| ⓖ | – | ✓ | ✓ | 65.73 | 78.98 | 30.34M | 15.77GB |
| ⓗ | ✓ | – | ✓ | – | – | 3.17M | ≫24GB |
| ⓘ with $\lambda_j$ | ✓ | ✓ | ✓ | 65.27 | 78.73 | 3.21M | 14.51GB |
| ⓙ ALL U-KAN | ✓ | ✓ | ✓ | **69.43** | **81.82** | 3.17M | 4.44GB |

Since our spline basis computation does not require storing gradients during training, this study adopts a chunk-wise computation strategy to further reduce memory consumption, as shown in Table 3. Notably, smaller chunk only increases runtime modestly while significantly reducing memory usage without affecting training performance. By default, the chunk size is set to 32 for input resolutions of $256 \times 256$ and 16 for $512 \times 512$.

Table 3: The relationship between chunk size, memory usage, and training speed. Batch size is 8 on the BUSI dataset.

| Method | Chunk | GPU Mem. | Time/step |
|---|---|---|---|
| U-KAN$_{MLP}$ | – | 2.49GB | 0.24s |
| U-KAN | – | 2.52GB | 0.27s |
| Ours | 8 | 3.82GB | 2.27s |
| Ours | 32 | 4.44GB | 1.37s |
| Ours | 64 | 6.20GB | 1.22s |
| Ours | 512 | 12.59GB | 1.14s |

## 5 CONCLUSION

This paper breaks the computational barriers that have prevented KANs from scaling to deeper, laying a solid foundation for future research in deep learning. (1) Based on Sprecher's variant of the KA theorem, we propose the SaKAN, which reduces training difficulty and enhances learning capacity. (2) Our Grad-Free Spline greatly reduces computational cost by detaching the spline basis gradients, enabling the practical deployment of deeper KANs. (3) Building on these innovations, we present the ALL U-KAN, which is is the first representative implementation of fully KA-based deep model. Experiments on three medical image segmentation tasks demonstrate that KA-based layers can fully substitute FC and Conv layers in deep learning while exhibiting superior performance. Although training KA-based models remains slower than traditional models due to spline computations, the speed is still acceptable in practice. Future research may focus on accelerating spline computations, which are independent of training because Grad-Free Spline detaches its gradient, to further improve training efficiency.

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

## A  LLM USAGE STATEMENT

We used a large language model (LLM) solely for grammar correction and minor language polishing of the manuscript. The LLM did not contribute to research ideation, methodology, analysis, or the generation of scientific content. All scientific contributions are entirely those of the authors.

