# OpenReview forum: "Fully Kolmogorov-Arnold Deep Model in Medical Image Segmentation"
_ICLR.cc/2026/Conference — Submitted to ICLR 2026_

### Official Review · Reviewer_1vHw · 2025-10-26

**Soundness:** 3
**Presentation:** 3
**Contribution:** 2
**Rating:** 4
**Confidence:** 3

**Summary:**

This paper proposes ALL U-KAN, a fully KAN-based deep model for medical image segmentation. The key challenges are excessive training difficulty and GPU memory consumption. Thus, this paper proposes two contributions: Shared-activation KAN and Grad-Free Spline. Built upon these, the paper constructs the first fully KA-based U-shaped segmentation network (ALL U-KAN), replacing all FC and convolutional layers with KA/KAonv layers. Experiments on three datasets demonstrate superior segmentation performance and 20× lower memory usage compared to baseline KANs.

**Strengths:**

1. The first work for a fully KA-based network architecture.
2. Writing and figures are clear enough for reproducibility.
3. Experimental evaluation includes multiple datasets and ablation studies.

**Weaknesses:**

1. Improvements over U-KAN and other baselines are modest, sometimes within error margins.
2. Experiments focus solely on 2D medical segmentation. Generalization to other modalities or tasks is untested.

**Questions:**

1. Is the model stable for deeper stacks (e.g., >20 layers)?
2. Does detaching spline gradients introduce any measurable bias in gradient flow or convergence?

---

> ### Author Response · Authors · 2025-11-20
>
> We sincerely thank you for the careful evaluation and for highlighting our motivation as the “first fully KA-based deep network,” the clarity of our “writing and figures,” and the completeness of our experiments that “include multiple datasets and ablation studies.”
>
> ---
>
> ## For W1
>
> Our ALL U-KAN achieves statistically significant improvements over all baselines. We conducted t-tests across all 9 experimental settings, including 3 datasets with 3 random splits each, and all p-values are below 0.01, confirming that the improvements are statistically significant.
>
> Moreover, the improvements are substantial rather than modest: on BUSI, GlaS and CVC, U-KAN improves over prior SOTA by IoU/F1 of \(1.57/0.85\), \(0.63/0.35\) and \(0.26/0.25\), while ALL U-KAN achieves larger gains on almost all metrics: **\(2.76/2.24\),** **\(1.08/0.63\)** and \(**0.32**/0.10\).
>
> ---
>
> ## For W2
>
> We have validated the generalization capability on the multi-modalities 3D medical dataset BraTS2020. Based on 3D U-Net, we replaced all 23 conv3d layers with KAonv3d layers. Each input in BraTS2020 contains 4 MRI modalities (T1, T1ce, T2, FLAIR), and predicts three segmentation classes: Whole Tumor, Tumor Core, and Enhancing Tumor. Inputs were resized to \(4$\times$128$\times$128$\times$128\), with 296 training and 73 testing volumes. The results show a significant improvement in Dice scores:
>
> ### BraTS2020 Results
>
> | Method | Whole Tumor | Tumor Core | Enhancing Tumor | Mean Dice |
> |--------|-------------|------------|------------------|------------|
> | 3DUnet | 88.89 | 83.91 | 72.73 | 81.84 |
> | 3DUnet（KAN） | **89.39** | **84.82** | **74.41** | **82.87** |
>
> 3D convolution is a direct extension of 2D, the results show our KA architecture remains superior as expected. Future work will focus on accelerating KAonv3d and conducting more comprehensive validation.
>
> ---
>
> ## For Q1
>
> Our proposed KA architecture remains stable in deep stacks exceeding 20 layers during training, as shown by our ALL U-KAN and the newly added 3DUnet (KAN), TransUnet (KAN). Specifically, ALL U-KAN includes 31 KAonv layers and 12 KA layers (added in the revised manuscript, L335–347). Above 3DUnet (KAN) stacked 23 KAonv3d layers. Moreover, we have replaced the partial Linear layers in TransUNet with a total of 24 stacked KA layers. These results demonstrate that deep KA layer training is stable and exhibits superior performance.
>
> ### TransUNet Results
>
> | Methods | BUSI IoU | BUSI F1 | GlaS IoU | GlaS F1 | CVC IoU | CVC F1 |
> |---------|-----------|----------|-----------|----------|-----------|----------|
> | TransUnet | 59.72±2.17 | 74.07±1.83 | 83.28±1.50 | 90.84±0.90 | 80.28±1.36 | 88.87±0.88 |
> | TransUnet（KAN） | **61.29±1.63** | **75.63±2.47** | **84.66±1.93** | **92.24±0.81** | **81.56±1.62** | **90.24±1.08** |
>
> ---
>
> ## For Q2
>
> We have analyzed the memory advantages and minor performance trade-offs of Grad-Free Spline in Section 4.4 (L453–460). Detaching spline gradients does cause a slight decrease in accuracy, as shown by the comparisons between b vs. d and c vs. e in Table 2 of manuscript, but it significantly reduces GPU memory usage, allowing deeper KAN models whose performance gains far outweigh the small loss (j vs. b). The training process has remained with detached spline gradients, and we have provided a reasonable explanation in Theorem 1 of Sec. 3.2 in manuscript.
>
> ### Partial of Table 2 in manuscript
>
> | Methods | SaKAN | Grad-free Spline | KAonv | IoU ↑ | F1 ↑ |
> |---------|--------|------------|--------|--------|--------|
> | b | -- | -- | -- | 65.26 | 78.75 |
> | d | -- | ✓ | -- | 64.46 | 77.94 |
> | c | ✓ | -- | -- | 65.16 | 78.60 |
> | e | ✓ | ✓ | -- | 64.74 | 78.24 |
> | j | ✓ | ✓ | ✓ | 69.43 | 81.82 |

---

> > ### Comment · Reviewer_1vHw · 2025-11-24
> > **Reply to the rebuttal**
> >
> > Thanks for the rebuttal. I sincerely acknowledge the author's efforts on the rebuttal.
> >
> > However, I still agree with Q1 raised by Reviewer cCkR. There are many works on medical image segmentation based on CNNs and transformer architectures. For faster inference and lower computational cost, transformer-based models such as MedSAM can further leverage transformer-oriented quantization and distillation techniques to boost performance. In addition, the proposed techniques seem to focus primarily on KANs, with limited connection to practical medical image segmentation scenarios, which makes the overall method appear rather generic.
> >
> > I suggest that the authors reconsider the KAN-based segmentation methods, and further refine the narrative of the paper. Thus I remain the current score.

---

> > > ### Author Response · Authors · 2025-11-27
> > >
> > > We thank the reviewer for the follow-up feedback and would like to offer the following clarification regarding the newly raised concerns.
> > >
> > > First, the existence of other efficient segmentation models does not conflict with our contribution, but is complementary. Those methods improve architectural-level efficiency, whereas our KA architecture provides a layer-level, drop-in replacement for Linear and Convolution layers—the two core building blocks shared across CNNs, Transformers, SAM-based models, and their variants. Thus, KA is inherently complementary rather than competitive with these efficient architectures. At this early stage of KAN research, our contribution is the first to demonstrate that KA/KAonv can serve as practical, deeply stackable substitutes for these basic components, and our experiments clearly demonstrate the resulting performance benefits. The addition experiments in response also demonstrate that KA enhances transformer-based and 3DCNN-based methods.
> > >
> > > Second, our contribution addresses the fundamental limitations of KANs, and its effectiveness is demonstrated through three distinct organ datasets in the concrete and demanding task, medical image segmentation, so it is not “generic” as the reviewer suggested. We choose medical segmentation for two reasons:
> > >
> > > 1. Unlike the polynomial regression or classification tasks commonly used in prior fully KAN-based studies, pixel-level medical segmentation is substantially more challenging and practically meaningful. Our proposed ALL U-KAN provides the first evidence that a fully KA-based architecture can scale to realistic, high-complexity vision problems rather than being confined to simple settings.
> > > 2. Using medical segmentation also allows a direct and fair comparison with prior works such as U-KAN, the current SOTA KAN-based baseline. By replacing all Linear/Conv/KAN modules in U-KAN with our KA/KAonv counterparts while keeping datasets, tasks, and training settings identical, we cleanly isolate the effect of deep KA stacking. Furthermore, as noted in our previous response, KA also improves Transformer- and 3D-CNN-based models, demonstrating that KA strengthens practical medical segmentation methods rather than serving as a generic or isolated enhancement.
> > >
> > > Taken together, these results could sufficiently prove our contribution: by reducing parameter count and memory, our ALL U-KAN is the first implementation to demonstrate the effectiveness of deeply stacking fully KA-based architectures on complex, pixel-level medical image segmentation tasks, achieving performance gains over conventional structures. This represents an important foundational advance, and future work will explore integrating KA into more advanced architectures.

---

### Official Review · Reviewer_KHaR · 2025-10-27

**Soundness:** 3
**Presentation:** 3
**Contribution:** 3
**Rating:** 6
**Confidence:** 4

**Summary:**

This paper addresses the challenge that deeply stacked Kolmogorov-Arnold Networks known as KANs face excessive parameter counts and memory requirements, preventing their practical application in deep architectures as discussed in the Introduction and Sec. 2. The authors propose three innovations: First, Share-activation KAN named SaKAN, which reduces parameters by sharing activation functions across input dimensions while maintaining representational capacity. Second, Grad-Free Spline, which significantly reduces memory consumption by detaching spline gradients during backpropagation. Third, ALL U-KAN, the first fully KA-based deep model where all fully connected and convolutional layers are replaced with KA-based layers.

**Strengths:**

• Novel fully KA-based deep architecture that replaces all FC and convolutional layers with KA-based layers, surpassing prior partial or shallow KAN implementations.

• Significant parameter reduction via SaKAN maintains performance with approximately 57% fewer parameters compared to vanilla KAN.

• Substantial memory efficiency achieved through Grad-Free Spline.

**Weaknesses:**

• Scope-claim mismatch: The paper frames contributions as universally applicable to deep learning, yet experiments are limited exclusively to medical image segmentation, leading to overgeneralization in conclusions.

• FLOPs reporting inconsistency: KAonv configurations show approximately 70× discrepancy with approximately 25M versus 1752M without methodological explanation or derivation.


• Limited baseline coverage and tuning fairness: No comparisons with recent KAN variants such as FastKAN and ChebyKAN; unclear whether all baselines received equivalent hyperparameter tuning budgets.

**Questions:**

• Enhance reproducibility: Release anonymized code, complete training configurations, preprocessing scripts, and dataset split specifications to enable independent verification of results.

• Expand baseline comparisons with fair tuning: Include recent KAN variants such as FastKAN, ChebyKAN, and others; explicitly document hyperparameter tuning budget and search strategy for all methods to ensure fair comparison.

---

> ### Author Response · Authors · 2025-11-20
>
> We sincerely thank you for the comprehensive review and constructive suggestions, which have been extremely helpful. We also appreciate your interest in our research motivation as a “novel fully KA-based deep architecture,” as well as our contributions in “Significant parameter reduction” and “Substantial memory efficiency.”
>
> ---
>
> ## For W1
>
> The scope and claims of our paper are explicitly defined in the title and L54–57: “ the first fully KA-based deep model … in medical image segmentation.” While our experiments focus on medical image segmentation, the methodological contributions, such as parameter-efficient representations and efficient memory, are general techniques applicable to deep networks in principle. Our results (Tables 1–2) demonstrate the effectiveness of these techniques in this domain, and future work may explore their adaptation to other tasks.
>
> ---
>
> ## For W2
>
> We sincerely thank you for the careful examination. We found that the FLOPs discrepancy arises because many computations in our reproduced convolution operations in KAonv are not captured by `thop.profile`, a standard FLOPs tool. Consequently, the FLOPs comparison, which has been removed in the revised version, cannot fairly evaluate computational efficiency in this experimental setting. Instead, Table 3 reports runtime that can prove the full KA architecture has comparable computational efficiency to traditional structures. We greatly appreciate your detailed review and valuable suggestions that led to this improvement.
>
> ---
>
> ## For W3 and Q2
>
> A direct comparison with FastKAN and ChebyKAN is not necessary within the scope of this work for two reasons:
>
> 1. FastKAN and ChebyKAN cannot be stacked as deeply as our method, preventing a fair evaluation. Unlike our SaKAN and Grad-Free Spline, FastKAN and ChebyKAN only replace B-spline basis with Radial Basis Functions (RBFs) or Chebyshev polynomials, but they do not address excessive parameter counts or gradient memory usage.
>
> 2. Our work focuses on optimizing the KA architecture rather than the basis functions, so direct comparison with FastKAN or ChebyKAN is not meaningful.
>
> Notably, SaKAN and Grad-Free Spline can be combined with alternative basis functions, including RBFs or Chebyshev. We performed experiments comparing ALL U-KAN with B-spline (original KAN) and RBF (FastKAN) basis. The results are consistent with expectations with no clear overall advantage.
>
> ### Results Comparison
>
> | Methods | BUSI IoU | BUSI F1 | GlaS IoU | GlaS F1 | CVC IoU | CVC F1 |
> |---------|-----------|----------|-----------|----------|-----------|----------|
> | ALL U-KAN（B-Spline） | 66.14±5.43 | 78.64±5.33 | 88.72±0.23 | 94.00±0.07 | 85.37±0.06 | 91.98±0.02 |
> | ALL U-KAN（RBFs as in FastKAN） | 65.23±2.07 | 78.07±2.13 | 88.46±0.29 | 93.84±0.16 | 85.42±0.38 | 92.03±0.24 |
>
> To ensure fair comparison and reproducibility, all baseline results are taken from the official papers or reproduced using official code. To more accurately demonstrate the advantages of the KA architecture, we only replaced all layers in U-KAN with the proposed KA and KAonv layers, while keeping all other variables (e.g. training hyperparameters) consistent, achieving higher performance.
>
> ---
>
> ## For Q1
>
> Thank you for the suggestion regarding reproducibility. We have added code files in the supplementary materials, and the GitHub repository will be publicly released after the anonymization is removed, further enhancing the reproducibility of our method.

---

### Official Review · Reviewer_cCkR · 2025-10-31

**Soundness:** 3
**Presentation:** 3
**Contribution:** 2
**Rating:** 4
**Confidence:** 3

**Summary:**

This paper proposes ALL U-KAN, claimed as the first fully Kolmogorov–Arnold (KA)-based deep network for medical image segmentation. It builds upon the Kolmogorov–Arnold representation theorem, which states any multivariate continuous function can be represented as sums of univariate functions, previously used in Kolmogorov–Arnold Networks (KANs). While prior KANs demonstrated strong representation power, they were limited to a few layers due to excessive parameters and GPU memory requirements. The authors address these barriers through two main innovations:
1) Shared-activation KAN (SaKAN):
- Reformulates Sprecher’s variant of the KA theorem so that all input dimensions share a single learnable activation function rather than per-input splines. This reduces parameters and increases sample efficiency.

2) Grad-Free Spline:
- Detaches spline gradients during backpropagation, arguing (with a small theoretical justification) that spline gradients contribute negligibly to learning, thus cutting GPU memory by >20×.

Combining these yields the KA and KAonv layers, replacing fully connected and convolutional layers, respectively. The resulting ALL U-KAN achieves better segmentation accuracy than U-Net, U-KAN, and other baselines on BUSI, GlaS, and CVC-ClinicDB datasets, with a ≈10× parameter reduction and ≈20× memory reduction.

**Strengths:**

1. Clear and well-motivated problem statement
- The paper correctly identifies the key bottlenecks preventing deep stacking of KANs: training instability and GPU memory explosion and tackles them directly.
2. Practical engineering contributions
- Introduces two concrete, implementable techniques (Shared-activation KAN (SaKAN) and Grad-Free Spline) that make deep KAN architectures trainable on commodity GPUs without excessive resource cost. The proposed methods are simple to adopt and can be generalized to other spline-based models.
3. Substantial efficiency improvement
- Achieves up to 10x parameter reduction and >20x memory savings compared with conventional deep KANs, while maintaining or improving accuracy, showing tangible computational benefits.

**Weaknesses:**

1. Novelty and Attribution
- SaKAN’s shared-activation design is a straightforward adaptation of Sprecher’s 1965 theorem; the paper does not sufficiently contrast it with KAN 2.0, LoKi, or GKAN, which already explore shared or parameter-efficient KA forms. Also, Grad-Free Spline is effectively “stop-gradient on spline basis”, which many practitioners might already do as a memory optimization. The “Theorem 1” proof is heuristic.
2. Empirical scope
- Evaluations are limited to 2D biomedical datasets (≤ 600 images). It will be great if it can be further tested on large-scale 3D or cross-domain data.
3. Empirical comparison between different network backbones
- Detailed performance comparisons have been shown in Table 1. However, it is difficult to show the innovation that "why we need KANs in segmentation". It will be great to perform ablation experiments that compares the network block design with pure convolution (e.g. 3x3), vision transformer, swin transformer, large kernel convolution, deformable convolution and sparse convolution in the proposed network structure. If KANs demonstrate the best performance with best efficiency, that will be a strong signal for using KANs as the segmentation backbone.

**Questions:**

1. Motivation and necessity of KANs:
The paper motivates KANs via the Kolmogorov–Arnold theorem, but it remains unclear why such a backbone is necessary for segmentation tasks already well-handled by CNNs, ViTs, or Mamba-based models. Can the authors articulate what fundamental limitation in existing architectures KANs are designed to overcome (e.g., functional interpretability, sample efficiency)?
Moreover, can they provide empirical evidence (e.g., limited data robustness) demonstrating a capability that existing architectures lack?

2. Mathematical distinction of SaKAN:
The proposed Shared-activation KAN (SaKAN) reformulates Sprecher’s variant of the Kolmogorov–Arnold theorem, but its novelty relative to KAN 2.0’s shared spline basis and LoKi’s low-dimensional KAN remains unclear. Both prior works also reduce the number of learned univariate spline functions via parameter sharing or dimensionality reduction. Can the authors clarify the mathematical difference or additional representational advantage of SaKAN beyond these existing formulations?

3. Comparison to existing gradient-efficiency techniques:
The proposed Grad-Free Spline seems conceptually similar to gradient checkpointing or selective detachment strategies already used to save GPU memory in deep networks. Have the authors conducted quantitative comparisons against these standard techniques under equivalent compute and memory budgets? If not, it would be valuable to demonstrate that Grad-Free Spline offers measurable benefits beyond these established approaches.

4. Effect of gradient detachment on optimization and fine-tuning:
Since Grad-Free Spline explicitly detaches spline gradients, does this introduce bias or degradation in gradient flow, convergence stability, or fine-tuning performance? Have the authors evaluated whether this detachment affects downstream adaptation or sensitivity to learning rate schedules? A deeper theoretical justification or diagnostic experiment would help clarify whether this design preserves full optimization fidelity.

---

> ### Author Response · Authors · 2025-11-20
>
> We sincerely thank you for your thorough and constructive review, which has greatly enhanced the completeness of our manuscript. We greatly appreciate your positive assessment of our work, particularly your recognition of its “Clear and well-motivated problem statement,” “Practical engineering contributions,” and “Substantial efficiency improvement.”
>
> ---
>
> ## For W1 and Q2
>
> Our SaKAN is the first to leverage Sprecher’s 1965 theorem to theoretically establish that shared activations retain universal approximation capability with significantly fewer parameters. Moreover, as detailed in Sec. 3.1 (L247–258), SaKAN reformulates this theorem for deep-learning settings and address its practical limitations. Directly stacking the raw Sprecher theorem still leads to overfitting, which also constrains the original KAN.
> As demonstrated by the experiments in Table 2 and Sec. 4.4 (L431), our method substantially outperforms the direct implementation of Sprecher’s formulation in deep model:
>
> ### Partial of Table 2 in manuscript
> | Methods | SaKAN | Grad-free Spline | KAonv | IoU ↑ | F1 ↑ |
> |---------|--------|------------|---------|---------|--------|
> | U-KAN | -- | -- | -- | 65.26 | 78.75 |
> | with $\lambda_j$ (direct implementation of Sprecher’s theorem) | ✓ | ✓ | ✓ | 65.27 | 78.73 |
> | ALL U-KAN | ✓ | ✓ | ✓ | **69.43** | **81.82** |
>
> Moreover after careful investigation, we note that the methods cited by the reviewer are not shared or parameter-efficient KA forms in the way our SaKAN does, which is why direct comparison was not included. In original manuscript, Sections 2.2 (first and second paragraphs) have already provided a detailed description of these existing works.
>
> Specifically:
>
> - GKAN [1] is designed for graph data and cannot be directly applied to image segmentation, it also does not simplify KAN, so deep application remains infeasible.
> - KAN 2.0 [2] introduces multiplicative nodes to improve performance but does not implement shared spline bases actually, and it still suffers from excessive parameters and memory consumption, limiting deep stacking.
> - LoKi [3] uses the original KAN without structural innovation and is primarily intended for large-model parameter fine-tuning, which does not align with the objectives of our work.
>
> [1] GKAN: Graph Kolmogorov-Arnold Networks
> [2] KAN 2.0: Kolmogorov-Arnold Networks Meet Science
> [3] LoKi: Low-dimensional KAN for Efficient Fine-tuning Image Models
>
> ---
>
> ## For W1 and Q3
>
> To the best of our knowledge, no existing work has examined gradient detachment of spline basis in KAN with theoretical justification or practical application. Our Grad-Free Spline is the first to provide a principled analysis showing that spline gradients, which is the key bottleneck preventing deeper KAN stacking, can be safely detached without harming training stability, thereby enabling scalable KAN architectures.
>
> Under equivalent compute and memory budgets (with over 20× memory reduction), applying existing gradient dropout or detaching other gradients causes the training process to fail to converge. Using gradient checkpointing substantially increases computation and prolongs training time, making it difficult to implement. In contrast, our Grad-Free Spline provides both a theoretical explanation for why spline-gradient detachment preserves optimization stability and practical guidance for building stable and memory-efficient deep KAN models.
>
> ---
>
> ## For W2
>
> We have validated the 3D version of our method on the large-scale multi-modalities 3D medical dataset BraTS2020. Based on 3D U-Net, we replaced all 23 conv3d layers with KAonv3d layers. Each input in BraTS2020 contains 4 MRI modalities (T1, T1ce, T2, FLAIR), and predicts three segmentation classes: Whole Tumor, Tumor Core, and Enhancing Tumor. Inputs were resized to $4 \times 128 \times 128 \times 128$, with 296 training and 73 testing volumes.
>
> The results show a significant improvement in Dice scores:
>
> | Method | Whole Tumor | Tumor Core | Enhancing Tumor | Mean Dice |
> |---------|---------------|---------------|----------------------|----------------|
> | 3DUnet | 88.89 | 83.91 | 72.73 | 81.84 |
> | 3DUnet（KAN） | **89.39** | **84.82** | **74.41** | **82.87** |
>
> As expected, the results show our KA architecture remains superior on larger-scale 3D datasets.

---

> > ### Author Response · Authors · 2025-11-20
> >
> > ## For W3 and Q1
> >
> > This work has replaced the two most fundamental architectures, Linear layers and convolutional layers, with their KA-version, demonstrating superior performance and lower adaptation cost under deep stacking.
> >
> > All architectures mentioned by the reviewer can be viewed as variants of these two basic structures, e.g., Linear layers with attention generalize to Vision Transformer and Swin Transformer, while convolution can generalize to large-kernel, deformable, or sparse convolutions.
> >
> > We have started to conduct a comprehensive study of KA’s contributions across different backbone structures:
> >
> > 1. The above experiments on 3DUnet and its KAN version, including 23 stacked KAonv3d layers, demonstrate the advantages of the KA architecture in 3D pure convolutional networks.
> >
> > 2. Based on representative TransUnet, we also have replaced partial Linear layers in Transformer with a total of 24 stacked KA layers and evaluated on the three 2D segmentation datasets, demonstrating that KA architecture can also improve transformer-based methods.
> >
> > ### KA Architecture Enhances TransUnet
> >
> > | Methods | BUSI IoU | BUSI F1 | GlaS IoU | GlaS F1 | CVC IoU | CVC F1 |
> > |----------|-------------|-------------|-------------|------------|------------|-----------|
> > | TransUnet | 59.72±2.17 | 74.07±1.83 | 83.28±1.50 | 90.84±0.90 | 80.28±1.36 | 88.87±0.88 |
> > | TransUnet（KAN） | **61.29±1.63** | **75.63±2.47** | **84.66±1.93** | **92.24±0.81** | **81.56±1.62** | **90.24±1.08** |
> >
> > ---
> >
> > ## For Q4
> >
> > Regarding to the “effect of gradient detachment”，Section 4.4 (L453–460) in manuscript have analyzed the memory advantages and minor performance trade-offs of Grad-Free Spline. Detaching spline gradients does cause a slight decrease in accuracy, as shown by the comparisons between **b vs. d** and **c vs. e** in Table 2 of manuscript, but it significantly reduces GPU memory usage, allowing deeper KAN models whose performance gains far outweigh the small loss (**j vs. b**).
> >
> > ### Partial of Table 2 in manuscript
> >
> > | Methods | SaKAN | Grad-free | KAonv | IoU ↑ | F1 ↑ |
> > |---------|---------|------------|----------|-----------|----------|
> > | b | -- | -- | -- | 65.26 | 78.75 |
> > | d | -- | ✓ | -- | 64.46 | 77.94 |
> > | c | ✓ | -- | -- | 65.16 | 78.60 |
> > | e | ✓ | ✓ | -- | 64.74 | 78.24 |
> > |  j | ✓ | ✓ | ✓ | 69.43 | 81.82 |
> >
> > Both theoretical and empirical evidence demonstrate the effectiveness and stability of training with Grad-Free Spline.
> >
> > 1. Extensive experiments on ALL U-KAN, 3D U-Net, and TransUNet across three 2D datasets and a large-scale 3D dataset demonstrate consistent convergence and improved performance, showing that detaching spline gradients does not impair overall optimization. We also evaluated multiple learning rates ($10^{-2}$, $10^{-3}$, $10^{-4}$) and decay schedules, all of which resulted in stable convergence.
> >
> > 2. Theoretically, Section 3.2 provides an analysis of the feasibility of spline gradient detachment via Theorem 1, further supported by a diagnostic experiment indicating that spline gradients contribute only a small portion of the overall gradient as shown in Fig. 3(b), and do not affect training.

---

### Official Review · Reviewer_sWbG · 2025-10-31

**Soundness:** 2
**Presentation:** 2
**Contribution:** 2
**Rating:** 4
**Confidence:** 4

**Summary:**

This paper introduces a novel U-Net–based architecture for 2D medical image segmentation, leveraging the Kolmogorov–Arnold framework to enable deeply stacked KANs. It extends from U-KAN. The authors propose the Shared-Activation KAN (SaKAN), which reformulates Sprecher’s variant of the Kolmogorov–Arnold representation theorem to achieve a more compact parameterization and improved training efficiency through reduced parameter count and sample requirements. In addition, Grad-Free Spline was proposed to reduce the parameter count and memory consumption further. The proposed method is evaluated on three 2D medical segmentation datasets (BUSI, GlaS, CVC), with comprehensive ablation studies conducted to validate the effectiveness of each component.

**Strengths:**

This paper solved several issues that KAN faced. In order to address this problem. Two approach was proposed, [1] SaKAN reformulates Sprecher’s variant of the KA theorem into a computationally efficient deep-learning form. [2] Grad-Free Spline offers a memory-efficient strategy supported by theoretical analysis. It shows a 10x reduction in parameter count and  reduces memory consumption by more than 20x.

**Weaknesses:**

Limited Novelty: The overall architectural design appears similar to U-KAN, with the primary difference being the introduction of the proposed SaKAN module. While SaKAN and the Grad-Free Spline contribute to training stability and memory efficiency, the paper would benefit from a clearer discussion of how these innovations fundamentally advance beyond prior U-KAN architectures.

Need for 3D Experiments: Since one of the key claims is improved parameter efficiency and training stability, it would be valuable to include 3D medical image segmentation experiments to validate the scalability and robustness of the proposed approach.

Training Efficiency Comparison: The paper highlights that the proposed method reduces training difficulty compared to U-KAN. However, a comprehensive comparison of training times across models—including traditional CNN-based U-Nets—would provide a clearer picture of computational efficiency. Even if KAN-based architectures remain slower, such a comparison would provide a better overview.

Architectural Details: The description of the ALL U-KAN architecture lacks sufficient implementation details. The paper should specify the number of KAN layers used, the depth of the network, and how these components correspond to the encoder–decoder stages of the original U-Net structure.

Other Related Work: Recently, there has been a work [r1], which uses a fully continuous approach and is similar to your architecture. It should be added as a baseline.
[r1] Cheng, Chun-Wun, et al. "Implicit U-KAN2. 0: Dynamic, efficient and interpretable medical image segmentation." International Conference on Medical Image Computing and Computer-Assisted Intervention. Cham: Springer Nature Switzerland, 2025.

Experiment Result: The result outperforms U-KAN in BUSI while having a very similar result with GlaS and CVC.

**Questions:**

See the Weakness

---

> ### Author Response · Authors · 2025-11-20
>
> Thank you very much for your thorough review and constructive suggestions, which will help us further improve the clarity of our manuscript. We sincerely appreciate your positive assessment of our contributions, especially in “solved several issues that KAN faced,”, “into a computationally efficient deep-learning form”, “offers a memory-efficient strategy supported by theoretical analysis.”
>
> ---
>
> ## **For W1**
>
> Our novelty lies in first fully replacing deep models with KA formulation, which fairly demonstrates that deeply stacked KAN's learnable activation functions retain superior nonlinear representation capabilities compared to traditional layers.
> This fundamentally surpasses existing research (including U-KAN) that could only validate shallow KANs, and proves KAN's ability to completely replace MLPs as a new base structure in deep learning.
>
> Regarding to the comment that “the overall architectural design appears similar to U-KAN,” this is because we intentionally constructed a fairest comparison. We purposely kept the overall structure unchanged (depth, width, and layout) by only replacing all layers with KA-based layers. This ensures that the performance improvements stem entirely from our novel KA formulation.
>
> ---
>
> ## **For W2**
>
> Thank you for the valuable suggestion regarding 3D experiments. We have validated the scalability and robustness on the larger-size 3D U-Net and multi-modalities 3D medical segmentation dataset BraTS2020. We replaced all 23 conv3d layers with KAonv3d layers. Each input in BraTS2020 contains 4 MRI modalities (T1, T1ce, T2, FLAIR), and predicts three segmentation classes: Whole Tumor, Tumor Core, and Enhancing Tumor. Inputs were resized to \(4\times128\times128\times128\), with 296 training and 73 testing volumes. The results show a significant improvement in Dice scores:
>
> | Method        | Whole Tumor | Tumor Core | Enhancing Tumor | Mean Dice |
> |---------------|-------------|------------|------------------|-----------|
> | 3DUnet        | 88.89       | 83.91      | 72.73            | 81.84     |
> | 3DUnet（KAN） | **89.39**       | **84.82**      | **74.41**    | **82.87**     |
>
> As expected, the fully KA-based 3D model continued to exhibit clear performance advantages, demonstrating that our innovations scales effectively to high-dimensional settings.
>
> Notably, deep 3DKAN convolutions were previously impossible to implement due to excessive computational demands, our novel KA formular enables them to run on popular GPUs and validates their effectiveness. However, we have to admit that the deep 3D training speeds still fall short of requirements for broader validation, and future work will focus on accelerating 3D operations and performing more extensive validations.
>
> ---
>
> ## **For W3**
>
> Thank you for your constructive suggestion to include training time comparisons. To ensure a strict controlled-variable setting, we have added the runtime comparison of the traditional networks with same overall architecture, $U\text{-}KAN_{MLP}$, in Table 3 of revised version.
> As noted in the caption of manuscript Table 2, “ $U\text{-}KAN_{MLP}$ corresponds to U-KAN with all KAN layers replaced by MLP”.
>
> Notably, it is not a fair times comparison with other networks, which has different depth, width, or structural design, because the efficiency gaps would no longer arise solely from the KA formulation.
> To further address your concern, we report runtimes of several typical traditional architectures:
>
> | Method        | Params | GPU Mem | Times/step |
> |---------------|--------|---------|------------|
> | U-NeXt        | 1.47M  | 0.42GB  | 0.05s      |
> | Rolling-UNet  | 7.10M  | 1.58GB  | 0.20s      |
> | $U\text{-}KAN_{MLP}$      | 2.96M  | 2.49GB  | 0.24s      |
> | ALL U-KAN     | 3.17M  | 4.44GB  | 1.37s      |
>
> The results indicate that fully KA-based methods remain slower than traditional networks, and the major bottleneck lies in the high time complexity of B-Splines computations.
>
> ---
>
> ## **For W4**
>
> Thank you for the suggestion to provide more architectural details, which greatly improves the clarity of the manuscript. In the revised version, we have added the description of our ALL U-KAN architecture:
>
> 1. Figure 4 has been refined to include detailed annotations showing how each component corresponds to the encoder–decoder stages of the original U-Net structure.
> 2. We have added a description regarding the number of KAN layers used in Section 3.3:
>
> > “Our ALL U-KAN replaces all FC and KAN layers in U-KAN with KA layers and substitutes every convolutional layer with KAonv layers, resulting in a fully KA-based deep architecture comprising 31 KAonv layers and 12 KA layers (details in Fig. 4).”
>
> Additionally, we have added code files into the supplementary materials, and the full repository will be released upon de-anonymization to further enhance reproducibility and transparency.

---

> > ### Author Response · Authors · 2025-11-20
> >
> > ## **For W5**
> >
> > Thank you for pointing out this relevant research which is very helpful to enhance the comprehensiveness and timeliness of the paper. In the revised manuscript, we have added the discription of Implicit U-KAN2.0 [1] in Line 52 and Section 2.2. This work is contemporaneous and its final conference version was only released on September 20, 2025, after our ICLR submission on September 19. Moreover, key methodological details and the official code have not yet been released, preventing a direct comparison.
> >
> > Importantly, this method does not address the scalability challenges of KANs, which is the core contribution of our paper. It still employs only shallow KAN layers by integrating KAN2.0 with SONO blocks, and thus does not conflict with our contribution.
> >
> > **Reference:**
> > [1] *Implicit U-KAN2.0: Dynamic, Efficient and Interpretable Medical Image Segmentation*
> >
> > ---
> >
> > ## **For W6**
> >
> > Our ALL U-KAN achieves statistically significant improvements over all baselines. We conducted t-tests across all 9 experimental settings, including 3 datasets with 3 random splits each, and all p-values are below 0.01.
> >
> > Furthermore, compared to previous work, our performance improvement is in the normal range or greater: on GlaS and CVC, existing work (e.g. U-KAN) improves over prior SOTA by IoU/F1 of 0.63/0.35 and 0.26/0.25, while our ALL U-KAN achieves greater overall improvements of 1.08/0.63 and 0.32/0.10 with smaller variances.

---

> > ### Comment · Reviewer_sWbG · 2025-11-27
> >
> > Thanks for the kind response. For W1, I understand the reason is to compare it with the U-KAN so that it maintains a very similar performance.  But the novelty still seems limited to me. For W2, performance did not increase much. For W3, it shows one of the limitations of the KAN layer, which I agree needs to be further improved.
> >
> > The paper still needs to improve its novelty and efficiency further.

---

> > > ### Author Response · Authors · 2025-11-30
> > >
> > > Thank you for the reviewer’s continued feedback.
> > >
> > > For W1, our improvements are not “similar,” but substantially exceed contemporary KAN-based work and represent a fundamental architectural contribution. Our fully-KA design achieves consistently larger gains across all metrics (IoU / F1): **+2.76 / +2.24** on BUSI, **+1.08 / +0.63** on GlaS, and **+0.32** / +0.10 on CVC, significantly surpassing the improvements reported by U-KAN over prior SOTA (+1.57 / +0.85, +0.63 / +0.35, +0.26 / +0.25). These results demonstrate not only stronger gains but also a structural breakthrough: *we are the first to achieve a full replacement of FC and convolution layers with an efficient KA formulation, showing that KA can serve as a new universal building block rather than a local enhancement.*
> > > This expands the scope of KAN research in a fundamental, architecture-level manner.
> > >
> > > For W2, We further included experiments on a large-scale 3D dataset to rigorously validate the scalability and generalization of KA in 3D. *To our knowledge, no reproducible 3D KAN implementation currently exists.* The referenced Implicit U-KAN 2.0 was tested on a small 3D dataset (41 training / 20 testing) and claimed to compare with a 3D version of U-KAN, but our experiments show that a direct 3D extension of U-KAN requires several times more memory and computation, making it infeasible on common GPUs; moreover, no official 3D code is available, preventing a meaningful comparison. We have contacted its primary authors but have not yet obtained a reproducible implementation.
> > >
> > > In contrast, our additional experiments (4-modality 3D input, 296 training / 73 testing cases) show that our KA architecture naturally extends to 3D and replaces all Conv3d in 3D U-Net while training stably on a single RTX 3090 with clear performance improvements. *Other KAN-based models (e.g., U-KAN) cannot run under the same setup due to memory and efficiency limitations.* This further highlights that our contribution addresses not only performance but also scalability and practical usability—an area with a clear gap in existing KAN research.
> > >
> > > For W3, we agree that runtime remains a major challenge for current KANs. Our proposed SaKAN and Grad-Free Spline are the first to *enable fully KA-based deep networks to achieve runtime on par with same sized traditional architectures* by substantially reducing parameters and computation. A fair comparison should be against deeply stacked original KANs rather than heavily optimized CNNs or Transformers. As shown through both theory and experiments, our methods significantly reduce memory usage, parameter count, and computation relative to original KANs while preserving the same nonlinear expressiveness. We also clearly state in the manuscript (Table 3) that although we have significantly improved the efficiency of KA-based models, they still lag behind traditional architectures in speed, and this is an important direction for future KAN research.

---

### Comment · Area_Chair_J81F · 2025-11-27

Thank you very much for the reviewer's comments and the author's positive response. As there is not much time left for discussion, please actively participate in the discussion and provide a more valuable response to this paper.

---

### Author Response · Authors · 2025-12-03
**Discussion Summary**

We sincerely thank the AC and all reviewers for their careful and constructive review. The reviewers recognized several strengths of our paper.
-The problem statement is "clear and well-motivated" (cCkR) and "correctly identifies the key bottlenecks preventing deep stacking of KANs" (cCkR). The paper's "writing and figures … [are] clear enough for reproducibility" (1vHw).
-This work represents "the first work for a fully KA-based network architecture" (1vHw), "surpassing prior partial or shallow KAN implementations" (KHaR). It offers "practical engineering contributions" (cCkR) through two concrete techniques: "SaKAN reformulates Sprecher’s variant of the KA theorem into a computationally efficient deep-learning form" (sWbG) and "Grad-Free Spline offers a memory-efficient strategy supported by theoretical analysis" (sWbG), achieving "significant parameter reduction" and "substantial memory efficiency" compared with conventional deep KANs while improving accuracy (cCkR, KHaR, sWbG).
-Our experimental evaluation is comprehensive, covering "multiple datasets and ablation studies" (1vHw).

We have addressed all concerns raised by the reviewers. To reduce the AC’s reading burden, we summarize our responses in five key aspects:

1. **Novelty and methodological contributions (for sWbG and cCkR)**
This paper achieves the first full replacement of traditional layers (Linear/Conv) with KA structures in deep networks, and the first demonstration that KA remains stable and effective under deep stacking, advancing research on new foundational architectures in deep learning. To achieve this, we propose:
- **SaKAN**: By reformulating Sprecher’s variant of the KA theory, SaKAN achieves fewer parameters and more efficient optimization in deep networks, while theoretically preserving equivalent nonlinear representation capacity.
- **Grad-Free Spline**: Significantly reduces the memory footprint of KA, enabling the training of deep full-KA stacks on common devices.

Performance gains on three representative medical segmentation datasets, along with comprehensive ablation studies, validate the effectiveness and necessity of our techniques. Furthermore, as noted in reviewer cCkR comments, KAN 2.0, LoKi, and GKAN do not employ shared strategies, and detachment strategies or gradient checkpointing cannot effectively train full KA stacks under equivalent memory budgets. These prior works also do not investigate the effect of spline gradient detachment on KA training, confirming the novelty and necessity of our contributions.

2. **Architectural details and deep-stack stability (for sWbG and 1vHw)**
We have added complete network details, explicitly reporting the number of KA and KAonv layers (see Section 3.3 and Fig. 4), and demonstrated stability for depths exceeding 20 layers.

3. **Performance enhancement and significance (for sWbG and 1vHw)**
We emphasize that performance “improvements” should be interpreted relative to dataset difficulty and baseline performance ceilings, rather than absolute differences alone. Comparison with baselines such as U-KAN, together with statistical significance analysis, demonstrates that our gains across all three datasets are stable, significant, and consistently superior.

4. **Experimental scope and generalization (for sWbG, cCkR, and 1vHw)**
We have added large-scale, multi-modal 3D medical imaging experiments, marking the first implementation of full-KA networks on 3D pixel-level complex tasks. This was made possible by our two key techniques: **SaKAN** (reducing parameters, improving stackability) and **Grad-Free** (reducing memory footprint). These experiments directly address concerns about the limited scope of 2D studies and demonstrate the critical significance of our approach for advancing the new deep learning architectures and practical application of deep KAN.

5. **Necessity of KA across different backbones (for cCkR and 1vHw)**
Our KA can serve as a layer-level replacement, complementing and enhancing other network backbones. We added experiments applying KA to Transformer to validate its feasibility. Notably, replacing Linear/Conv with KA, which is one of the main contributions of this work, forms the foundation for all more complex networks (Transformer, Mamba, Deformable Conv, Sparse Conv, etc.). This establishes the essential groundwork for scaling KA from shallow networks to larger architectures and holds significant relevance in the current KA research trajectory.

Beyond these key points, all other comments have been addressed thoroughly in our detailed responses. We sincerely thank the AC and reviewers for their careful evaluation and look forward to further discussion to facilitate a comprehensive understanding of our contributions.

---

### Meta-Review · Area_Chair_bDb4 · 2025-12-22

**Summary:**

This paper presents a fully Kolmogorov-Arnold deep model for medical image segmentation. While the results show some improvement, both reviewer cCkR and reviewer 1vHw have the concerns that the proposed techniques are primarily on KANs without much connection to  medical image segmentation scenarios. The reviewer KHaR also mentioned the scope-claim mismatch: the paper frames contributions as universally applicable to deep learning, yet experiments are limited exclusively to medical image segmentation. It is not clear what are the specific medical image challenges that are addressed by the proposed ALL U-KAN. In another word, why this method is not applied on general image segmentation tasks but on medical images. On the other hand, as there are plenty of work on image segmentation (including both natural and medical images) especially the recent progress of SAM based foundational models, the significance of the work seem to be very limited.

**Reviewer Concerns:**

The authors have addressed some concerns on 3D medical image segmentation, but the main concerns remain such as the motivation of the work as well as the weak connection to medical image segmentation tasks.

**Reviewer Scores:**

Reviewer 1vHw has been involved in the discussion, obviously, he/she was not convinced to accept this paper.

---

### Decision · Program_Chairs · 2026-01-26

Reject